# Quality of life among leprosy patients: An analysis of associated factors

Alo Edin[1]*, Ziyad Towfik[2], Dejene Tesfaye[2], Abduselam Asefa[2], Abdulmalik Abdela Bushra[3], Ibsa Abdusemed Ahmed[4], Angefa Ayele[1]

**1** Department of Epidemiology, Institute of Health, Bule Hora University, Bule Hora, Ethiopia, **2** Department of Psychiatry, Collage of Health and Medical Sciences, Haramaya University, Haramaya, Ethiopia, **3** East Hararghe Zonal Health Department, Oromia Regional State, Harar, Ethiopia, **4** School of Public Health, College of Health and Medical Sciences, Haramaya University, Harar, Ethiopia

* jiblosa1@gmail.com

## Abstract

Leprosy is accompanied by stigma and discrimination that significantly impact patients' quality of life. Previous studies in Ethiopia have focused primarily on demographic and disease-related aspects, neglecting mental health factors. Therefore, was aimed to assess quality of life and associated factors among leprosy patients in Eastern Ethiopia. An institutional-based cross-sectional study was conducted among 170 leprosy patients at Bisidimo General Hospital using the World Health Organization Quality of Life-BREF (WHOQOL-BREF) questionnaire. Data were analyzed using STATA version 17, with multiple linear regression identifying associations between variables. Mean domain scores were: physical 53.03 ± 16.44, psychological 51.37 ± 13.10, social 44.95 ± 18.34, and environmental 57.10 ± 15.28. Depression, anxiety, comorbidity, stigma, and illness duration showed significant negative associations with quality of life domains. Income demonstrated positive associations across all domains. Additionally, poor social support negatively impacted the physical domain, while lifetime khat use negatively affected the social domain. Leprosy patients exhibited lower scores in social and psychological domains but higher scores in the environmental domain. Most factors affecting quality of life are preventable, suggesting the importance of programs that increase awareness, facilitate early treatment, and implement routine screening for mental health alongside physical health to improve patients' quality of life.

## Introduction

Leprosy or Hansen's disease is a chronic granulomatous infectious disease of the skin and peripheral nerves, caused by M. leprae and the more recently discovered M. Lepromatosis [1–3]. It is considered to be a neglected disease, being endemic in poor countries where the detection rates show only a slight trend toward a decrease [4,5]. Despite

**Data availability statement:** All necessary information were included with the paper and its Supporting Information files.

**Funding:** The author(s) received no specific funding for this work.

**Competing interests:** The authors have declared that no competing interests exist.

the global decrease in leprosy prevalence, Ethiopia remains one of the countries with a high burden of the disease [6]. Quality of life (QoL) refers to an individual's perspective of his or her circumstances within the various social, cultural, and value systems in which they reside. Quality of life addresses various aspects of a person's life span, such as financial security, fulfillment of personal expectations and goals, psychological and social welfare, and physical, emotional, and mental health [7]. Stigma and discrimination which is associated with Leprosy can significantly impact the quality of life (QoL) of affected individuals [8]. The psychological health of patients living with leprosy is affected by several economic and social factors and intimately connected with QoL [9].

The presence of pain, discomfort, and reduced sensation further diminishes the quality of life experienced by individuals. The psychological and emotional well-being of patients with leprosy is often marred by social stigma and discrimination [10]. Throughout history, leprosy has been associated with a notable social stigma, leading to discrimination, exclusion, and isolation due to misconceptions surrounding the disease. The stigma associated with leprosy can lead to self-stigma, where individuals internalize negative societal attitudes, further exacerbating their mental health challenges. This self-stigma may manifest as low self-esteem, decreased self-efficacy, and reluctance to seek medical or psychological help [9,11–13].

Study conducted in Africa showed that, 14% and 34.6% of patients with leprosy had a depression and anxiety [14]. In Ethiopia, the prevalence of mental disorders ranged from 47.4% to 52% among patients with leprosy [15], Which results in reduced social engagement and a decline in the quality of life for patients [10]. The stigma encircling leprosy can impose a substantial burden and impact various aspects of an individual's quality of life and mental well-being [16]. Leprosy affects physical well-being through nerve damage, skin lesions, and deformities, limiting mobility and function [8]. It also leads to psychological and emotional distress, including depression and anxiety [10]. Social stigma and discrimination contribute to mental health issues, while the fear of contagion can lead to social isolation [13]. The disease also results in a loss of livelihood, as physical impairments can hinder employment and social engagement. Overall, leprosy negatively impacts individuals' overall quality of life and their ability to function effectively [11]. Besides poor quality of life has devastating consequences, not only for their individual well-being, but also impacts families, and society [17,18]. Furthermore, poor QoL can lead to treatment compliance issues, deformities, adjustment problems, unemployment.

The lack of study on the correlation between mental health factors and quality of life in leprosy patients is a critical gap that hinders the development of holistic treatment strategies. Existing studies predominantly conducted in high-income countries have focused on socio-demographic and disease-related aspects, overlooking the profound stigma and discrimination faced by individuals with leprosy. This study area in resource-limited settings lacks comprehensive research on quality of life in leprosy patients. While previous studies have used various assessment measures, only a few have utilized the WHOQOL-Bref standardized tool [19–23] to evaluate individuals affected by leprosy and its influencing factors. Therefore, this study aims to fill this gap by assessing the quality of life and its associated factors among leprosy patients in Eastern Ethiopia.

## Methods

### Ethics statement

To ensure human subjects, ethical clearance for the research work was obtained from the Institutional Health Research Ethics Review Committee (IHRERC) College of Health and Medical Science of Haramaya University with a reference number of Refw-IHRERC/141/2024. Then, the obtained formal permission letter was taken to BGH before the data collection. Before data collection the objective of the study, potential risk and benefits, confidentiality and privacy of the study was clearly explained to the participants and, informed, voluntary participation, written and signed consent was obtained. Confidentiality was ensured throughout the accomplishment of the study. Participants was informed that their participation is voluntary that they can withdraw from the study at any time if they wish to do so. All the information given by the participants were used for research purposes only. Besides, to keep the anonymity of study participants, code numbers rather than personal identifiers was used. The research procedure was performed in compliance with the Declaration of Helsinki of the World Medical Association.

The study was conducted in Bisidimo General Hospital from June 15 to July 15, 2024. It is located in the Erer valley, about 22Km to the east of Harar town at the end of a turn-off from the road to Jigjiga. It is established in 1958 by German Leprosy and Tuberculosis Relief Association (GLRA). The hospital has 4 major departments and two speciality clinics. It around-10,000 outpatients visits have been documented in the 2023. A total of 175 patients were attended leprosy OPD during the study period.

### Study design

An institutional-based cross-sectional study design was conducted.

### Population

The study population comprised all patients with leprosy aged ≥18 at BGH who visited the leprosy OPD and were present during the data collection period. Patients diagnosed with leprosy who presented with concurrent skin diseases, or physical disabilities unrelated to leprosy were excluded from the study.

### Sample size and sampling procedure

The sample size for this study was calculated using the single population mean formula, with the following parameters: a 95% Confidence Level (Z(a/2) = 1.96), a standard deviation of 17.13 ($\sigma = 17.13$), a margin of error of 5% of the mean (d = 2.15) [24], and accounting for a 10% non-response rate. However, we have included all 170 consecutive leprosy patients who met the eligibility criteria during the data collection and received care at Bisidimo General Hospital.

According to the 3rd quarter report of 2023, the total number of adult patients attending the leprosy outpatients department (OPD) at Bisidimo General Hospital during the quarter was 540, with an average monthly OPD attendance of 180 patients. However, the number of patients specifically diagnosed with leprosy was relatively small. Out of the 175 leprosy patients who visited the leprosy OPD at Bisidimo General Hospital during the study period, 170 patients met the eligibility criteria and were consecutively recruited for the study. This consecutive sampling approach ensured that all eligible participants who attended the OPD during the data collection period were included, maximizing coverage and representation of the target population.

### Data collection methods and tools

An interviewer-administered structured questionnaire was used to collect the data about the characteristics of patients with leprosy such as; Socio demographic characteristics (age, sex, marital status, educational status, occupation, monthly income), clinical factors; (Leprosy disability Grade, duration of the illness and comorbidity obtained from patients card (document review)), Quality of life of patients with leprosy was assessed by using the WHOQOL-BREF-26 items

questionnaire. The questionnaire consists of two parts. The first, part assesses the individual's overall perceptions of quality of life and the person's overall perception of health. The second part evaluates the four domains: physical health (7 items), psychological health (6 items), social relationships (3 items), and environmental health (8 items). These four domain scores denote individual's perception of life in a particular domain. Each of these items scored from 1 to 5 on a response scale. Domain scores scaled in a positive direction (i.e., higher scores correspond to a better quality of life). The QoL raw scores are transformed into a range between 0 and 100. The mean scores of the items with in each group's domain used to calculate the domain score. A mean score is the multiplied by 4 to make domain scores comparable with scores used in WHOQOL-100 which produce score range 4–20. The second transformation method converts domain scores to a scale 0–100 [25].

PHQ-9: To screen depression, the Patients Health Questionnaire (PHQ-9) was used. It is a 9-point item questionnaire and every item incorporate a 4-point Likert scale that ranges from 0 ("not at all") to 3 ("nearly every day"), generating a complete score starting from 0 to 27. Moreover, PHQ-9 has been validated in Ethiopian healthcare context with specificity and sensitivity of 67 and 86% respectively. A cut-off points of 10 and above has been used to screen depression [26,27].

GAD-7: To screen anxiety, the Generalized Anxiety Disorder (GAD-7) was used. It is a 7- point item questionnaire and every item incorporates a 4-point Likert scale that ranges from 0 ("not at all") to 3 ("nearly every day"), generating primary a complete score starting from 0–21. The scale presents a rapid, efficient, reliable, and valid method for detecting the presence of anxiety disorder. A cut-off points at a score of 10 and above on the GAD-7 scale had been defined as anxiety with a sensitivity and specificity of 89% and 82% respectively [28–31].

ASSIST: The Alcohol, Smoking, and Substance Involvement Screening Test (ASSIST) was used to assess substance use-related problems. It was developed under the auspices of the World Health organization by an international group of addiction researchers and clinicians and it has 8 items. Ever substance use is the use of substance during life time and Current substance use is the use of substance since the last 3 months [15].

EMIS-AP: The questionnaire was used to assess perceived stigma among patients with leprosy. The EMIS-AP includes 15 questions. Each stigma item was rated on a four-point scale, 'No' (0), 'Don't know' (1), possible (2), and Yes (3), concerning stigma. The overall stigma score using the EMIS-AP scale is calculated by summing the responses to the relevant items. The total score reflects the perceived stigma experienced by individuals related to leprosy. Higher scores indicate greater perceived stigma [32,33].

OSSS-3: The OSSS-3 is a self-reported social supported scale. It consists of 3 items with a total score ranging from 3–14. Scores from 3 to 8 was considered to indicate poor support, scores from 9 to 11 indicate intermediate support and a score between 12 and 14 is considered to indicate strong social support [34].

Suicide Behavior Questionnaire-Revised (SBQ-R); The first item was used to assess. inquired whether the respondent had ever had thoughts of suicide or engaged in suicidal behavior in his/her lifetime. Item 1 question 2 was utilized for suicidal ideation throughout one's life, item 1 question 3a and 3b for suicide plans, and item 1 question 4a and 4b for suicidal attempts, And validated in Nigeria in clinical and non-clinical settings with Cronbach's alpha of 0.80, the score ranged 0–3 and cutoff point ≥1 indicate suicidality.

Data were collected by four mental health professional specialist who had experience and trained on data collection, and one supervisors supervised the data collection process. To assure data quality, the team was trained and oriented for one day regarding the tools, objectives, methods of data collection, and other related ethical issues.

## Data collection procedure

Data was collected using structured questionnaire with face-to-face interview. Before the data collection, patients with leprosy who meet the eligibility criteria were given an informed, voluntary and written consent form to sign after being read to them about the study 's goals, objectives, and purpose of the study. Then, the data collectors interviewed those participants qualified for the study at convenient locations, while supervisor oversaw the data collection process.

## Variables

**Dependent variable.** Quality of life.

**Independent variables.** Socio demographic: (Age, Sex, Educational status, Occupational status, Income, and Marital status). Clinical factors: (Leprosy disability Grade, duration of the illness and comorbidity was obtained from patient's card). Psychosocial related factors: (Perceived Stigma and social support). Mental health related factors: (Depression, Anxiety, suicidal attempt and Substance use factors: Alcohol, cannabis, Khat, or/and Tobacco use).

## Operational definitions

**Quality of life:** Assessed by WHO-BREF-26; Scores range from 0-100 on WHO-BREF with high score close to 100 indicates higher quality of life and low score closer to 0 indicates lower quality of life [35].

**Anxiety:** Individual who score on GAD-7; Anxiety (≥10) indicative of Anxiety, None (0–4), Mild (5–9), Moderate (10–14) and Severe (> 15) [36].

**Depression:** Individual who score on PHQ-9; Major depression (≥10) indicative, none (0–4), Mild (5–9), Moderate (10–14) and Severe (15–27) [37].

**Substance use:** was considered on ASSIST if participant use of alcohol, cigarette, khat, etc., once or more.

**Lifetime substance use**: was considered if participant use of alcohol, cigarette, khat, etc., once or more in the participant's lifetime.

**Current substance use:** Use of substance mentioned in the lifetime substance use once or more during the past three months [38].

**Perceived Stigma:** Individual who score on EMIC-AP; high score indicates higher Perceived Stigma [39].

Low stigma: a total score below median score on EMIC-AP.

High stigma: a total score above median score on EMIC-AP.

**Social support:** Individual who score on OSS-3: 12–14 indicate Strong social support, 9–11 indicate Intermediate social support and 3–8 indicate Poor social support [34].

## Data quality control

To ensure the data quality, pre-test was conducted on 5% (10) of the sample population before the actual data collection at Haramaya General Hospital. Training was given for data collectors and supervisors about research objectives, data collection tools, and procedures for one day. The principal investigator with one supervisors, were supervise the technique of data collection and completeness of tools on the daily basis to give appropriate feedback accordingly.

## Methods of data analysis

Following data collection, the questionnaire underwent thorough checks for completeness and consistency before being entered into EPI Data version 3.1 for analysis using STATA version 17 statistical software. Descriptive statistics were utilized to generate summary statistics including mean, standard deviation, frequency, and percentage. Simple linear regression analysis was conducted to select candidate variable at $P < 0.25$. Variables at this level were then considered as candidates for multiple linear regression to identify independent predictors of quality of life, after validating the assumptions of linear regression. Normality of residuals was confirmed through skewness and kurtosis tests, with a skewness range between -1 and 1 indicating normal distribution. Heteroscedasticity was assessed using the Breusch-Pagan/Cook-Weisberg test, with no significant findings across all domains. Potential outliers were assessed using cook distance, with no influential outliers detected based on a cut-off point of ±1. Variance inflation factor (VIF) was used to assess multicollinearity among independent variables, with a VIF cut-off point below 5 indicating no multicollinearity concerns. Statistical significance of independent predictors was determined at a 95% confidence level ($P < 0.05$), with β coefficients

used for interpretation. Variables with a P-value less than 0.05 in the multiple linear regression analysis were considered statistically significant.

## Ethics statement

To ensure human subjects, ethical clearance for the research work was obtained from the Institutional Health Research Ethics Review Committee (IHRERC) College of Health and Medical Science of Haramaya University with a reference number of Refw-IHRERC/141/2024. Then, the obtained formal permission letter was taken to BGH before the data collection. Before data collection the objective of the study, potential risk and benefits, confidentiality and privacy of the study was clearly explained to the participants and, informed, voluntary participation, written and signed consent was obtained. Confidentiality was ensured throughout the accomplishment of the study. Participants was informed that their participation is voluntary that they can withdraw from the study at any time if they wish to do so. All the information given by the participants were used for research purposes only. Besides, to keep the anonymity of study participants, code numbers rather than personal identifiers was used. The research procedure was performed in compliance with the Declaration of Helsinki of the World Medical Association.

## Results

### Socio-demographic characteristics

A total of 170 patients with leprosy participated in the study, achieving a response rate of 100%. Among these patients, two-thirds (65.29%) were male. The median age of the respondents was 34.5, with an interquartile range (IQR) of 18 (median: 34.5; IQR: 25–43). In terms of education, one-third (32.36%) were unable to read and write, while 32.94% attended primary school. The majority of participants (55.88%) were single, with a quarter (25.29%) being married. Regarding occupation, nearly one-third (30%) reported agriculture as their primary occupation, while 31.76% identified as housewives [Table 1].

### Clinical and psycho-social characteristics

Among the 170 study participants, nearly half (43.53%) reported poor social support, while 20.59% had a comorbid condition. Additionally, almost half (49.41%) experienced grade-0 disability. The median duration of illness for participants was 3 years, with an interquartile range (IQR) of 3 (median: 3; IQR: 2–5). The mean score for perceived stigma among participants was 31 (±10.7), with a majority (51.76%) reporting low perceived stigma [Table 2].

### Mental health and substance related characteristics

From the total participants, approximately one-third 53(31.18%) exhibited symptoms of depression, with a majority (48.82%) showing mild indications of depression. Additionally, one-third (29.41%) displayed symptoms of anxiety, while a majority (62.94%) had mild anxiety symptoms. Suicidal ideation was reported by 12.35% of the participants. In terms of substance use, 18.24% reported ever using alcohol, with 1.76% using alcohol in the past three months. Furthermore, 56.47% reported ever using cigarettes, and 22.35% used cigarettes in the past three months. Additionally, approximately two-thirds (62.35%) reported ever using khat, while one-third (34.12%) used khat in the past three months [Table 3].

In this study, the majority of participants, accounting for 62%, were lifetime khat users, followed by 34% who were current khat users. Additionally, 56% reported lifetime tobacco smoking, with alcohol being the least consumed substance among the participants (**Fig 1**).

### Mean score of overall and each domain of quality of life

In this study, the overall perception of quality of life (QOL) was reported to have a mean score of 2.70±1.00, with a 95% confidence interval ranging from 2.55 to 2.85. The overall perception of health scores had a mean of 2.35±1.04, with a

**Table 1. Socio-demographic characteristics of Patients with leprosy attending Bisidimo General Hospital Eastern Ethiopia, 2024 (n = 170).**

| Variable | Category | Frequency | Percentage |
|---|---|---|---|
| Age (median & IQR) | 34.5(25-43) | | |
| Household Income (median & IQR) | 2000 (1200-2500) | | |
| Sex | Male | 111 | 65.29 |
| | Female | 59 | 34.71 |
| Educational status | Unable to read and write Primary school Secondary school College and above | 55 | 32.36 |
| | | 56 | 32.94 |
| | | 44 | 25.88 |
| | | 15 | 8.82 |
| Marital Status | Married | 43 | 25.29 |
| | Single | 95 | 55.88 |
| | Separated/Divorced | 24 | 14.12 |
| | Widowed | 8 | 4.71 |
| Occupation | Agriculture | 51 | 30.00 |
| | Employee | 14 | 8.24 |
| | Business | 26 | 15.29 |
| | housewife | 54 | 31.76 |
| | others* | 25 | 14.71 |

**Table 2. Psychosocial and clinical related characteristics of patients with leprosy attending Bisidimo General Hospital Eastern Ethiopia, 2024 (n = 170).**

| Variable | Category | Frequency | Percentage |
|---|---|---|---|
| Social Support | Poor | 74 | 43.53 |
| | Moderate | 58 | 34.12 |
| | Good | 38 | 22.35 |
| Comorbid condition | Yes | 35 | 20.59 |
| | No | 135 | 79.41 |
| Disability Grade | Grade 0 | 84 | 49.41 |
| | Grade 1 | 66 | 38.83 |
| | Grade 2 | 20 | 11.76 |
| Perceived Stigma (mean & SD) | 31 ± 10.7 | | |
| Illness Duration(median, IQR) (in year) | 3 ± 2 | | |

95% confidence interval between 2.19 and 2.51. The mean score for the environmental domain was notably higher at 57.10 ± 15.28, with a 95% confidence interval from 54.78 to 59.41. On the other hand, the social domain had the lowest mean score at 44.95 ± 18.34, with a 95% confidence interval ranging from 42.17 to 47.73. These findings suggest varying levels of perceived quality of life across different domains, with environmental aspects being rated more positively compared to social aspects [Table 4].

### Self-rating quality of life and self-reported health satisfaction

The majority of the subjects expressed a negative subjective assessment of their quality of life 65(37.65%), particularly in terms of satisfaction with their health condition. The most common responses were neither satisfied nor dissatisfied with

Table 3. Mental health and substance related factors among patients with leprosy attending Bisidimo General Hospital, Eastern Ethiopia, 2024 (n = 170).

| Variable | Category | Frequency | Percentage |
|---|---|---|---|
| Depression | Yes | 53 | 31.18 |
| | No | 117 | 68.82 |
| None | Yes | 34 | 20.00 |
| Mild | Yes | 83 | 48.82 |
| Moderate | Yes | 31 | 18.24 |
| Severe | Yes | 22 | 12.94 |
| Anxiety | Yes | 50 | 29.41 |
| | No | 120 | 70.59 |
| None | Yes | 13 | 7.65 |
| Mild | Yes | 107 | 62.94 |
| moderate | Yes | 31 | 18.24 |
| Severe | Yes | 19 | 11.18 |
| Suicidal ideation | Yes | 21 | 12.35 |
| | No | 149 | 87.65 |

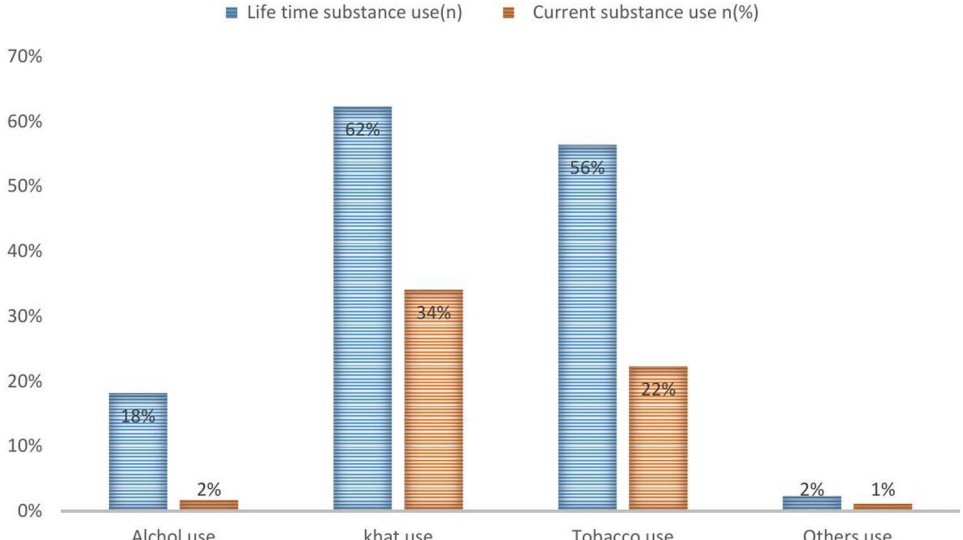

Fig 1. Life time and Current substance use among Patients with leprosy attending Bisidimo General Hospital, Eastern Ethiopia, 2024 (n =170).

their health condition, followed by being dissatisfied with their health condition. This indicates a prevalent sense of ambivalence or dissatisfaction among the subjects regarding their current health status [Table 5].

## Factors associated with quality of life among patients with leprosy

Linear regression analyses for physical domain of quality of life among Patients with leprosy.

In the simple linear regression analysis, 13 out of the 22 variables were identified as candidates for multiple linear regression at a significance level of p < 0.25. However, in the final multiple linear regression model, 7 variables were found

**Table 4. Mean score of quality of life among leprosy patients attending Bisidimo General Hospital, Eastern Ethiopia, 2024 (N = 170).**

| Variables | Mean (± SD) | SE | 95% CI |
|---|---|---|---|
| Q1-item | 2.70(±1.00) | 0.07 | (2.55-2.85) |
| Q2-item | 2.35(±1.04) | 0.08 | (2.19- 2.51) |
| Physical domain | 53.03 (±16.44) | 1.26 | (50.54-55.514) |
| Psychological domain | 51.37 (±13.10) | 1.04 | (49.39- 53.36) |
| Social domain | 44.95 (±18.34) | 1.41 | (42.17-47.73) |
| Environmental domain | 57.10 (±15.28) | 1.02 | (54.78-59.41) |

**Table 5. Assessment of the quality of life and health satisfaction among patients with leprosy attending Bisidimo General Hospital, Eastern Ethiopia, 2024 (n = 170).**

| Variable | Item category | Frequency(n) | Percent (%) |
|---|---|---|---|
| Perceived quality of life | Very poor | 17 | 10.00 |
| | Poor | 64 | 37.65 |
| | Neither poor nor good | 45 | 26.47 |
| | Good | 41 | 24.12 |
| | Very good | 3 | 1.76 |
| Perceived health satisfaction | Very dissatisfied | 45 | 26.47 |
| | Dissatisfied | 46 | 27.06 |
| | Neither dissatisfied nor satisfied | 57 | 33.53 |
| | Satisfied | 19 | 11.18 |
| | Very satisfied | 3 | 1.76 |

to be significantly associated with the physical domain of quality of life at $p < 0.05$. For example, the score of the physical domain of quality of life was lower among leprosy patients with depression ($\beta = -10.35$, 95% CI: -13.83, -6.86) compared to those without depression. Similarly, patients with anxiety had a lower score in the physical domain of quality of life ($\beta = -4.77$, 95% CI: -7.92, -1.93) compared to those without anxiety. Additionally, the physical domain score was lower among leprosy patients with comorbid diseases ($\beta = -4.20$, 95% CI: -7.57, -0.83) compared to those without comorbidities. Furthermore, each unit increase in the stigma score was associated with a decrease in the physical domain of quality of life ($\beta = -0.54$, 95% CI: -0.81, -0.28) among leprosy patients. An increase in illness duration was linked to a decrease in the physical domain score of quality of life ($\beta = -0.49$, 95% CI: -0.87, -0.11) among leprosy patients. Moreover, increases in income were associated with an increase in the physical domain score of quality of life ($\beta = 0.004$, 95% CI: 0.003, 0.005) among leprosy patients. The physical domain score was lower among those with poor social support ($\beta = -5.78$, 95% CI: -9.60, -1.96) and lower among leprosy patients with moderate social support ($\beta = -4.32$, 95% CI: -7.99, -0.65) compared to those with good social support [Table 6].

In the simple linear regression analysis conducted, out of the 22 variables assessed, 11 were identified as candidates for multiple linear regression with a significance level of $p < 0.25$. However, in the final multiple linear regression model, 5 variables were found to be significantly associated with the physical domain of quality of life at $p < 0.05$. For instance, the score of the psychological domain of the quality of life was lower among leprosy patients with depression ($\beta = -5.01$, 95% CI: -9.08, -0.95) compared to those without depression. Similarly, patients with anxiety exhibited a lower score in the psychological domain of quality of life ($\beta = -4.04$, 95% CI: -7.69, -0.39) compared to those without anxiety. Moreover, the psychological domain score was lower among leprosy patients with comorbid diseases ($\beta = -5.84$, 95% CI: -9.74, -1.94) compared to those without comorbid diseases. Additionally, an increase in the stigma score was associated with a

**Table 6. Linear regression analyses of factors associated with physical, domain quality of life among Patients with leprosy attending Bisidimo General Hospital, Eastern Ethiopia, 2024 (n = 170).**

| Variables | Physical domain | |
|---|---|---|
| | Simple Linear regression | Multiple Linear regression |
| | β Coef [95% CI] | β Coef [95% CI] |
| Sex | | |
| Male | -3.39(-8.37, 1.58) # | -0.39(-2.48, 3.27) |
| Female | 1 | 1 |
| Age (in Year) | -0.16(-.36, 0.05)# | 0.02(-0.09, 0.12) |
| Occupation | | |
| Agriculture | -16.12(-25.48, -6.75) ** | 1.00(-6.61, 4.59) |
| Merchant | -0.7.40(-17.69, 2.89)# | 0.257(-5.37, 5.88) |
| House wife | -17.65(-26.96, -8.34) *** | 0.50(-5.25, 6.25) |
| Other | -11.76(-6.76,2.71)* | -1.29(-6.90, 4.31) |
| Gov't employed | 1 | 1 |
| Income (in Birr) | 0.007(.006,.009)*** | 0.004(0.003, 0.005)*** |
| Comorbidity | | |
| Yes | -15.37(-21.09, -9.66)*** | -4.20(-7.57, -0.83)* |
| No | 1 | 1 |
| Disease duration | -1.67(-2.32, -1.01)*** | -0.49(-0.87, -0.11)* |
| Perceived Stigma | -1.77(-2.08, -1.47)*** | -0.54(-0.81, -0.28)*** |
| Social support | | |
| Poor | -23.32(-28.57, -18.06)*** | -5.78(-9.60, -1.96)** |
| Moderate | -7.66(-13.16 -2.16)** | -4.32(-7.99, -0.65) * |
| Good | 1 | 1 |
| Depression | | |
| Yes | -24.27(-28.19, -20.35)*** | -10.35(-13.83, -6.86)*** |
| No | 1 | 1 |
| Anxiety | | |
| Yes | -17.64(-22.42, -12.87)*** | -4.77(-7.92, -1.93)** |
| No | | |
| Suicidal ideation | | |
| Yes | -4.81(-12.36, 2.74)# | -2.24(-1.77, 6.26) |
| No | | |
| Life time Khat use | | |
| Yes | -5.30(-10.39, -.21)* | -1.60(-4.40, 1.19) |
| No | 1 | 1 |
| Life time alcohol use | | |
| Yes | -4.97(-11.39, 1.45)# | -1.95(-5.40, -1.50) |
| No | 1 | 1 |
| Current substance use | | |
| Yes | -12.45(-28.81, 3.91)# | -3.69(-12.07, 4.68) |
| No | 1 | 1 |

CI: confidence interval *p-value <0.05 **p-value <0.01 ***p-value <0.001.

Linear regression analyses for psychological domain of quality of life among Patients with leprosy.

**Table 7. Linear regression analyses of factors associated with psychological domain quality of life among patients with leprosy attending Bisidimo General Hospital, Babile, Oromia Region, Easter Ethiopia, 2024 (n=170).**

| Variables | Psychological domain | |
|---|---|---|
| | Simple Linear regression | Multiple Linear regression |
| | β Coef [95% CI] | β Coef [95% CI] |
| Occupation | | |
| Agriculture | -7.32(-15.20, 0.56) # | -0.69(-6.90, 5.52) |
| Merchant | 2.98(-5.68, 11.64) | 4.69(-1.81, 11.19) |
| House wife | -4.57(-12.41, 3.26) | 3.62(-2.79, 10.02) |
| Other | -0.71(-9.43, 8.00) | 0.85(-5.60, 7.30) |
| Gov't employed | 1 | 1 |
| Income (in EBirr) | 0.003(0.002, 0.004)*** | 0.002(-0.00, 0.003)* |
| Comorbidity | | |
| Yes | -15.72(-20.22, -11.22)*** | -5.84(-9.74, -1.94)** |
| No | 1 | 1 |
| Disease duration | -1.12(-1.67, -0.57)*** | -0.12(-0.56, 0.31) |
| Perceived Stigma | -1.14(-1.43, -0.85)*** | -0.47(-0.77, -0.16)** |
| Social support | | |
| Poor | -13.91(-18.47, -9.35) *** | -3.85(-8.28, 0.58) |
| Moderate | -2.33(-7.10, 2.43) | -2.64(-6.85, 1.56) |
| Good | 1 | 1 |
| Depression | | |
| Yes | -17.20(-20.80, -13.61)*** | -5.01(-9.08, -0.95)** |
| No | 1 | 1 |
| Anxiety | | |
| Yes | -14.15(-18.13, -10.17)*** | -4.04(-7.69, -0.39)* |
| No | | |
| Suicidal ideation | | |
| Yes | -6.39(-12.58, -0.20)* | -2.35(-7.12, 2.42) |
| No | | |
| Life time Khat use | | |
| Yes | -4.48(-8.68, -0.28)* | -2.91(-6.14, 0.32) |
| No | 1 | 1 |
| Life time alcohol use | | |
| Yes | -3.69(-9.00, -1.62)# | -2.43(-6.42, -1.57) |
| No | 1 | 1 |

CI: confidence interval *p-value <0.05 **p-value <0.01 ***p-value <0.001.

Linear regression analyses for social domain of quality of life among Patients with leprosy.

decrease in the psychological domain of quality of life (β=-0.47, 95% CI: -0.77, -0.16) among leprosy patients. Furthermore, an increase in income was linked to an increase in the psychological domain score of quality of life (β=0.002, 95% CI: 0.000, 0.003) among leprosy patients [Table 7].

In the simple linear regression analysis, out of the 22 variables examined, 12 were identified as candidates for multiple linear regression at a significance level of p<0.25. However, in the final multiple linear regression model, 5 variables were found to be significantly associated with the physical domain of quality of life at p<0.05. These variables included depression, anxiety, perceived stigma, comorbid disease, illness duration, income, and social support, all of which were significantly linked to all domains of quality of life.

For example, the score of the social domain of the quality of life was lower among leprosy patients with depression (β = -7.85, 95% CI: -12.61, -3.09) compared to those without depression. Similarly, patients with anxiety had a lower social domain score (β = -6.83, 95% CI: -11.12, -2.54) compared to those without anxiety. Additionally, each unit increase in the stigma score was associated with a decrease in the social domain of quality of life (β = -0.62, 95% CI: -0.98, -0.26) among leprosy patients. Furthermore, the social domain score was lower among leprosy patients who were lifetime Khat users (β = -3.92, 95% CI: -7.74, 0.11) compared to non-users. On the other hand, increases in income were associated with an increase in the social domain score (β = 0.005, 95% CI: 0.003, 0.006) among leprosy patients, while holding other variables constant [Table 8].

In the simple linear regression analysis, out of the 22 variables assessed, 13 were identified as candidates for multiple linear regression at a significance level of $p < 0.25$. However, in the final multiple linear regression model, 5 variables were found to be significantly associated with the physical domain of quality of life at $p < 0.05$. These variables included depression, anxiety, perceived stigma, comorbid disease, illness duration, income, and social support, all of which were significantly linked to all domains of quality of life.

For instance, the score of the environmental domain of quality of life was lower among leprosy patients with depression (β = -10.93, 95% CI: -14.85, -7.00) compared to those without depression. Similarly, patients with anxiety had a lower environmental domain score (β = -4.64, 95% CI: -8.17, -1.10) compared to those without anxiety. Moreover, each additional year of illness duration was associated with a decrease in the environmental domain score (β = -0.75, 95% CI: -1.17, -0.33) among leprosy patients.

Furthermore, an increase in the stigma score was linked to a decrease in the environmental domain score (β = -0.30, 95% CI: -0.60, -0.01) among leprosy patients. Conversely, an increase in income was associated with an increase in the environmental domain score (β = 0.003, 95% CI: 0.002, 0.004) among leprosy patients after adjusting for other variables [Table 9].

## Discussion

The study reported mean scores for the quality of life domains as follows: the physical domain had a mean score of 53.03 ± 16.44; the psychological domain had a mean score of 51.37 ± 13.10; the social domain had a mean score of 44.95 ± 18.34; and the environmental domain had a mean score of 57.10 ± 15.28. This study revealed that patients with leprosy scored highest in the environmental domain and lowest in the social and psychological domains of the WHOQOL-BREF. This finding aligns with studies conducted in Ethiopia among patients with podoconiosis who were assessed using a similar WHOQOL-BREF tool [40]. It is also consistent with research conducted in Brazil and China among leprosy patients, indicating higher scores in the environmental and physical domains compared to the social and psychological domains [41,42]. This suggests that patients may experience challenges in personal relationships, social support, and sexual activity, possibly due to the impact of leprosy on these aspects and the persistence of social exclusion and discrimination in Ethiopian communities despite awareness campaigns [43]. The higher environmental quality of life score implies that patients feel satisfied with safety, access to treatment, and financial aspects. This could be attributed to many leprosy patients in Ethiopia benefiting from vocational training programs and the expansion of healthcare services in rural areas [44,45].

Studies conducted in Ghana, Congo, Bangladesh, India, and Indonesia have reported the highest mean scores of quality of life in the social domain. The variations in quality of life scores among these countries could be attributed to cultural differences. In Ethiopia, for example, the cultural stigma associated with leprosy can result in social isolation and mental health challenges, significantly impacting the social and psychological domains of quality of life [43]. Additionally, when comparing the WHOQOL-BREF across different locations, local environment, social dynamics, and cultural factors may influence the results, highlighting the importance of considering these aspects in quality of life assessments [46].

**Table 8. Linear regression analyses of factors associated with social domain quality of life among patients with leprosy attending Bisidimo General Hospital, Eastern Ethiopia, 2024 (n = 170).**

| Variables | Social domain | |
|---|---|---|
| | Simple Linear regression | Multiple Linear regression |
| | β Coef [95% CI] | β Coef [95% CI] |
| Sex | | |
| Male | -4.03(-9.57, 1.52) | -0.35(-4.27, 3.57) |
| Female | 1 | 1 |
| Age (in Year) | -0.22(-0.44, 0.01) | 0.009(-0.15, 0.13) |
| Occupation | | |
| Agriculture | -22.51(-32.78, -12.24)*** | -4.62(-11.89, 2.66) |
| Merchant | -13.05(-24.33, 1.77)* | -4.23(-11.88, 3.42) |
| House wife | -23.57(-33.77, -13.36) *** | -2.82(-10.55, 4.91) |
| Other | -17.02(-28.39, -5.66)** | -4.86(-12.41, 2.69) |
| Gov't employed | 1 | 1 |
| Income (in EBirr) | 0.008(0.006,0.009)*** | 0.005(0.003,0.006)*** |
| Comorbidity | | |
| Yes | -13.73(-20.29, -7.17) *** | -2.78(-7.39, 1.82) |
| No | 1 | 1 |
| Disease duration | -1.77(-2.50, -1.03) *** | -0.45(-0.97, 0.07) |
| Perceived Stigma | -1.87(-2.23, -1.51) *** | -0.62(-0.98, -0.26)** |
| Social support | | |
| Poor | -22.99(-29.21, -16.77) *** | -4.07(-9.29, 1.14) |
| Moderate | -7.96(-14.46, -1.46) ** | -3.75(-8.76, 1.27) |
| Good | 1 | 1 |
| Depression | | |
| Yes | -23.51(-28.34, -18.67) *** | -7.85(-12.61, -3.09)** |
| No | 1 | 1 |
| Anxiety | | |
| Yes | -19.29(-24.65, -13.93) *** | -6.83(-11.12, -2.54)** |
| No | | |
| Life time Khat use | | |
| Yes | -7.56(-13.20, -1.93)** | -3.92(-7.74, 0.11)* |
| No | 1 | 1 |
| Life time alcohol use | | |
| Yes | -5.00(-12.28, -2.17) | -3.13(-7.84, -1.58) |
| No | 1 | 1 |
| Current substance | | |
| Yes | -17.76(-43.45, 7.93) | -2.07(-24.09, 19.94) |
| No | 1 | 1 |

CI: confidence interval *p-value <0.05 **p-value <0.01 ***p-value <0.001.

Linear regression analyses for environmental domain of quality of life among Patients with leprosy.

The findings of this study indicated lower levels of quality of life compared to studies conducted in Sri Lanka, India, and Indonesia across all dimensions [47,48]. This variance may be attributed to socio cultural factors specific, where cultural stigma associated with leprosy can result in social isolation and mental health challenges, significantly affecting

**Table 9. Linear regression analyses of factors associated with environmental domain quality of life among patients with leprosy attending Bisidimo General Hospital, Eastern Ethiopia, 2024 (n=170).**

| Variables | Environmental domain | |
|---|---|---|
| | **Simple Linear regression** | **Multiple Linear regression** |
| | **β Coef [95% CI]** | **β Coef [95% CI]** |
| Sex | | |
| Male | -3.33(-7.33, 0.68) | 2.34(-0.87, 5.55) |
| Female | 1 | 1 |
| Age (in Year) | -0.16(-0.33, -.01) | -0.03(-0.15, 0.09) |
| Marital status | | |
| Single | 2.48(-2.36, 7.31) | 1.14(-2.44, 4.73) |
| Separated | 0.25(-5.76, 6.251) | 3.32(-1.38, 8.01) |
| Widowed | -3.40(-13.08, 6.28) | 3.62(-3.69, 10.92) |
| Married | 1 | 1 |
| Occupation | | |
| Agriculture | -14.47(-21.90, -7.03) *** | -0.39(-6.39, 5.60) |
| Merchant | -6.95(-15.12, 1.22)# | -1.03(-7.33, 5.25) |
| House wife | -16.48(-23.88, -9.09) *** | -2.28(-8.64, 4.08) |
| Other | -11.54(-19.76, -.3.30)** | -1.85(-8.06, 4.37) |
| Gov't employed | 1 | 1 |
| Income (in EBirr) | 0.006(0.005, 0.007)*** | 0.003(0.002, 0.004)*** |
| Comorbidity | | |
| Yes | -12.57(-17.17, -7.97) *** | -2.84(-6.63, 0.95) |
| No | 1 | 1 |
| Disease duration | -1.56(-2.08, -1.05) *** | -0.75(-1.17, -0.33)** |
| Perceived Stigma | -1.37(-1.63, -1.11) *** | -0.30(-0.60, -0.01)** |
| Social support | | |
| Poor | -16.46(-21.62,-11.30)*** | -1.18(-5.45, 3.10) |
| Moderate | -0.63(-6.02, 4.77) | 2.01(-2.11, 6.13) |
| Good | 1 | 1 |
| Depression | | |
| Yes | -19.45(-22.64,-16.27) *** | -10.93(-14.85, -7.00)*** |
| No | 1 | 1 |
| Anxiety | | |
| Yes | -15.07(-18.85, -11.29) *** | -4.64(-8.17, -1.10)* |
| No | | |
| Life time Khat use | | |
| Yes | -5.08(-9.17, -0.99)* | -2.41(-5.34, 0.52) |
| No | 1 | 1 |
| Life time alcohol use | | |
| Yes | -10.81(-23.99, 2.39) | -6.24(-19.25, 6.78) |
| No | 1 | 1 |
| Current substance | | |
| Yes | -12.26(-30.86, 6.34) | -0.17(-18.30, 17.95) |
| No | 1 | 1 |

CI: confidence interval *p-value <0.05 **p-value <0.01 ***p-value <0.001.

the social and psychological domains of quality of life [43]. Additionally, socioeconomic factors play a crucial role, as economic disparities can influence access to resources, healthcare services, and education, thereby impacting overall quality of life [49].

However, the findings of this study is higher than studies conducted in Brazil, China, and Nepal across all dimensions of quality of life [41,42,50], and also outperform studies conducted in Ghana and Congo, except in the social domain [51,52]. This disparity may stem from variations in participant demographics, study settings, and many other factors. Unlike our study, the studies in Brazil focused on patients with neuropathic pain, while the study in China was community-based. Community-based studies have the advantage of capturing a broader spectrum of pain experiences that can impact quality of life [53]. In contrast, our study was conducted in a hospital setting at an outpatients department, where patients may have better access to treatments, psychological support, and other healthcare services that can enhance their overall well-being [54].

The study highlights a significant association between anxiety, depression, and reduced quality of life across all domains. These results are in line with studies carried out in Nepal and Egypt [50,55], indicating that individuals with poorer mental health tend to experience lower quality of life. This correlation can be attributed to the exacerbation of the disease and associated disabilities by depression and anxiety, leading to decreased motivation among patients to adhere to treatment plans. Mental health issues can significantly impact psychological well-being, resulting in social withdrawal and isolation, which in turn affects patients' living conditions [56].

The study revealed a significant negative correlation between perceived stigma and all domains of quality of life. This finding is consistent with several studies conducted in Congo, Nepal, and Indonesia [48,50,52,54], which highlighted the adverse impact of perceived stigma on the quality of life of leprosy patients across all domains. High levels of perceived stigma were found to have detrimental effects on the quality of life among these individuals, leading to delays in seeking treatment, heightened stress due to fear of judgment, social isolation to avoid discrimination, and potential residence in marginalized communities with limited healthcare access, ultimately diminishing their quality of life [57,58].

Furthermore, the study also indicated a significant association between income and all domains of quality of life. This result aligns with the studies conducted in India and Sri Lanka [47,49], which demonstrated that higher income levels are linked to better quality of life across all domains. The potential explanation for this association lies in the fact that higher income can afford individuals better living conditions, access to healthcare services, and proper nutrition, all of which contribute to an enhanced quality of life [46].

The study presented a significant association between poor social support and a decline in the physical domain of quality of life. This result is consistent with earlier studies conducted in Ethiopia among patients with NTD assessed using DLQI [15]. The potential explanation for this correlation lies in the fact that social support plays a crucial role in enhancing the self-care capabilities of patients with leprosy. When individuals receive support from their families, friends, and healthcare providers, they are more inclined to comply with treatment protocols, uphold proper hygiene practices, and participate in physical rehabilitation exercises, ultimately leading to improved physical well-being [53].

The study revealed a significant association between lifetime khat use and a decreased quality of life in the social domain. This finding is in line with a prior study conducted in Ethiopia among patients with NTD, indicating that substance use is linked to a lower quality of life [15]. This connection may be attributed to the fact that substance use can contribute to heightened social isolation and impaired relationships [59].

In this study, the presence of comorbid conditions was found to be significantly linked to lower levels of quality of life in both physical and psychological domains. This correlation aligns with previous studies carried out in Sri Lanka [49]. One possible explanation for this relationship is that comorbid conditions have the potential to worsen disabilities, increase stress and anxiety levels, ultimately diminishing the overall physical and psychological well-being of the patients.

The research findings indicate a significant negative correlation between illness duration and the physical and environmental domains of quality of life (QoL). This aligns with previous studies conducted in Egypt and India [47,56]. The

prolonged duration of illness may result in physical disabilities, complications, and comorbidities, making it challenging for patients to access adequate housing and healthcare [60,61].

### Strength and limitations of the study

As a strength this study has employed the WHOQOL-Bref standardized tool to assess individuals affected by leprosy and its associated factors. However, it is important to acknowledge the limitations of this study. Firstly, its cross-sectional nature prevents the establishment of a cause-and-effect relationship. Secondly, there is a possibility of informational bias, including recall and social desirability biases, especially concerning sensitive issues that may carry stigma. Lastly, the study's restriction to a single hospital limits its generalizability to all leprosy patients in the Oromia region.

## Conclusions

The findings indicate that the mean QoL scores were highest in the environmental domain ($57.10 \pm 15.28$) and physical domain ($53.03 \pm 16.44$), and lowest in the social domain ($44.95 \pm 18.34$) and psychological domain ($51.37 \pm 13.10$). Factors such as depression, anxiety, perceived stigma, comorbid diseases, illness duration, and lack of social support were linked to lower QoL, while income was associated with higher QoL. Health professionals are advised to conduct thorough assessments encompassing physical, psychological, and social aspects to promptly identify and address mental health issues in leprosy patients, thus enhancing their QoL. It is important to note that this study was limited to a clinical setting without a control group. Future research should consider including both clinical and community settings with control groups for a more comprehensive analysis.

## Acknowledgments

We are grateful to all contributors who generously shared their suggestions and insights with us. Special thanks are extended to the diligent data collectors for their invaluable support and assistance during the data collection phase.

## Author contributions

**Conceptualization:** Alo Edin, Ziyad Towfik, Dejene Tesfaye, Abduselam Asefa, Abdulmalik Abdela Bushra, Ibsa Abdusemed Ahmed, Angefa Ayele.

**Data curation:** Alo Edin, Ziyad Towfik, Dejene Tesfaye, Abduselam Asefa, Abdulmalik Abdela Bushra, Ibsa Abdusemed Ahmed, Angefa Ayele.

**Formal analysis:** Alo Edin, Ziyad Towfik, Dejene Tesfaye, Abduselam Asefa, Abdulmalik Abdela Bushra, Ibsa Abdusemed Ahmed, Angefa Ayele.

**Investigation:** Alo Edin, Ziyad Towfik, Dejene Tesfaye, Abduselam Asefa, Abdulmalik Abdela Bushra, Ibsa Abdusemed Ahmed, Angefa Ayele.

**Methodology:** Alo Edin, Ziyad Towfik, Dejene Tesfaye, Abduselam Asefa, Abdulmalik Abdela Bushra, Ibsa Abdusemed Ahmed, Angefa Ayele.

**Project administration:** Ziyad Towfik, Dejene Tesfaye, Abduselam Asefa.

**Resources:** Alo Edin, Ziyad Towfik.

**Software:** Alo Edin, Ziyad Towfik, Dejene Tesfaye, Abdulmalik Abdela Bushra, Ibsa Abdusemed Ahmed, Angefa Ayele.

**Supervision:** Ziyad Towfik, Dejene Tesfaye, Abduselam Asefa.

**Validation:** Alo Edin, Ziyad Towfik, Dejene Tesfaye, Abduselam Asefa, Abdulmalik Abdela Bushra, Ibsa Abdusemed Ahmed, Angefa Ayele.

**Visualization:** Alo Edin, Ziyad Towfik, Dejene Tesfaye, Abduselam Asefa, Abdulmalik Abdela Bushra, Ibsa Abdusemed Ahmed, Angefa Ayele.

**Writing – original draft:** Alo Edin, Ziyad Towfik, Dejene Tesfaye, Abduselam Asefa, Abdulmalik Abdela Bushra, Ibsa Abdusemed Ahmed, Angefa Ayele.

**Writing – review & editing:** Alo Edin, Ziyad Towfik, Dejene Tesfaye, Abduselam Asefa, Abdulmalik Abdela Bushra, Ibsa Abdusemed Ahmed, Angefa Ayele.

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
