## [Decision Letter · Decision Letter 0]

26 Feb 2025

PMEN-D-24-00585

Quality of Life among Leprosy Patients: An Analysis of associated Factors

PLOS Mental Health

Dear Dr. Edin,

Thank you for submitting your manuscript to PLOS Mental Health. After careful consideration, we feel that it has merit but does not fully meet PLOS Mental Health’s publication criteria as it currently stands. Therefore, we invite you to submit a revised version of the manuscript that addresses the points raised during the review process.

EDITOR:

Address all the concerns raised by the reviewers

We look forward to receiving your revised manuscript.

Kind regards,

Kizito Omona, PhD

Academic Editor

PLOS Mental Health

Journal Requirements:

1. We suggest you thoroughly copyedit your manuscript for language usage, spelling, and grammar. If you do not know anyone who can help you do this, you may wish to consider employing a professional scientific editing service.

Additional Editor Comments (if provided):

Address the concerns raised by the reviewers

Reviewers' comments:

Reviewer's Responses to Questions

**Comments to the Author**

1. Does this manuscript meet PLOS Mental Health’s publication criteria?

Reviewer #1: Yes

Reviewer #2: Yes

Reviewer #3: Yes

2. Has the statistical analysis been performed appropriately and rigorously?

Reviewer #1: Yes

Reviewer #2: I don't know

Reviewer #3: Yes

3. Have the authors made all data underlying the findings in their manuscript fully available (please refer to the Data Availability Statement at the start of the manuscript PDF file)?

Reviewer #1: Yes

Reviewer #2: Yes

Reviewer #3: Yes

4. Is the manuscript presented in an intelligible fashion and written in standard English?

Reviewer #1: Yes

Reviewer #2: Yes

Reviewer #3: No

Reviewer #1: I am pleased to write this recommendation regarding their recent publication .The work presented is not only thorough and well-researched but also offers valuable insights into the field of mental health and chronic infectious disorders

Throughout the paper, the authors demonstrates a strong command of the subject matter, backed by a solid methodology and clear presentation.

Reviewer #2: Abstract:

First sentence in line 21 doesnt add any significance informaton to the abstract.

Same applies to lines 26 and 27: "Consequently, there is limited information available on26 the quality of life of patients with leprosy in Ethiopia, particularly in Eastern Ethiopia."

In lines 28 to 28, the objective of the study does not match the literature gap (mental health) highlighted above.

From lines 32 to 37, the authors provided excess explanations that are not required in abstract. Instead mention the key activities. for instance, consider deleting information on "completeness and consistency", reasons for using linear regression ...

Introduction

Lines 74 and 75: Two consecutive sentences started with the word "despite". Consider revising it.

Be consistent with the use of abbreviations for Quality of Life. choose between QOL and QoL.

Lines 84 to 96, the ideas are scattered. Authors should organize their presentation well. for instance, they may start a description of the psychoclogical repercussion, then the causes (clinical or environmental) and manifestation in the lifes of the patients, etc....

Line 103: check grammatical error "which resulting..."

Line 104-105: "effects of stigma", has seem to be repeated in the Intro. Unless the authors intends to provide statistical evidence, they should consider reorganizing the flow of the writeup well.

Although the authors have provided some empirical evidence on mental health and leprosy, a more elaborate review is needed to determine the literature gap or context gap that this study is filling. Otherwise they should provide a stronger justification for the study.

Methods

Line 133: Authors should expalin the research design in the context of the study and give justifications. They should be mindful not to just give generic definitions from literature.

Line 134: Further description of the study population should be considered indicating ligibility criteria. Authors should explain why other skin diseases were condered and include this information in the backgroud. If the study is not just about leprosy, then the authors may consider revising the topic as well as the objective of the research.

Line 142: What is the eligibility criteria?

Line 138 to 143: How did the authors calculate the sample size?

Lines 149 to 159: What is the reliability of the WHOQOL-100 tool?

Lines 161 to 172: HQ9, GAD7, ASSIST, EMIS-AP and SBQ-R were not mentioned in the Abstract. Authors should consider correcting this omission.

Discussion

Well written discussion. Authors should consider referencing specific results in the dicsussion of the various themes. But be cautious not to present too many results which could result in repeatition of the results section.

Reviewer #3: Overall an interesting and important article. However, the manuscript is difficult to read and understand in large part due to a multitude of grammatical errors and awkward phrasing of sentences. Please consider the suggestions below to improve the quality of the paper and make it more suitable for publication.

- Line 76, insert comma between “curable” and “leprosy”

- Line 76, insert “is” between “leprosy” and “associated

- Line 76, change the “e” in discrimination to an “a”

- Lines 77-78, the word affected is used consecutively in the same sentence. Consider the following rephrase instead: the psychological health of patients living with leprosy is affected by several economic and social factors intimately connected with QoL.

- Regarding the introduction, for better flow of the paper, recommend describing “quality of life” prior to discussing how QoL is decreased in patients living with leprosy (i.e. place lines 79-83, before lines 74-78). This will better contextualize the argument of the paper for your readers.

- Line 85, it should say “patients” not “patient”; throughout the manuscript, the authors tend to say “patient” where “patients” instead should be used, please make this correction throughout the manuscript.

- Line 88, discuss further what these misconceptions are or provide examples of the most common misconceptions held about leprosy

- Recommend removing lines 88-89 or condensing with the earlier sentences (lines 86-88) as you are now repeating the same points without expanding further on them

- Line 89, patient should be the plural for, patients

- Recommend rephrasing lines 93-85. For example: “The social lives of patients with leprosy are often non-existent, rather these patients often endure isolation, primarily stemming…”

- Lines 97-98, consider the following rephrase: “patient with leprosy are 30% more likely to experience mental disorders globally, of which 70% experience depressive disorders while another 28% experience anxiety disorders”

- Line 100-102, this sentence as it is currently written is very confusing, consider rewriting entirely: Studies conducted in Nigeria have shown that of patients with leprosy, 14% have depression while 34.6% have anxiety.

- Line 103, change “leprosy of patients” to “patients with leprosy”

- Lines 103-104, the sentence that comes after “leprosy of patients” does not immediately make sense, what is the connection between this sentence and the one before it (lines 102-103), what is the conclusion/point being made here? Please rephrase for better clarification

- Line 113, insert a period between society and furthermore, as furthermore should be the start of a new sentence

- Line 128, change “is” to “was”

- Lines 129-130, the phrasing of this sentence is confusing, please rephrase for clarity

- Line 130, what does OPD stand for? Since this is the first mention of the acronym, the full name should be written out for readers

- Please explain the study design further. When institutions were apart of the project? Was there an IRB protocol that was approved by all institutions? This section should provide more detail than what is current presented in the manuscript

- Line 134, in the population section, please describe your inclusion and exclusion criteria further. Were all patients with leprosy accepted for the study regardless of age? Why were those with leprosy but other skin disease excluded from consideration?

- Line 155, this phrase is confusing and not needed “which agreed as a five-point liker scale?” Please either rephrase the sentence or remove

- Lines 175-176, when defining “ever substance use” and “current substance use,” please clarify if these are the definitions that the authors themselves came up with for the purposes of the study or if they are the definitions associated with the ASSIST screen

- Line 179, change EMIC-AP to EMIS-AP

- Line 194, consider this rephrase instead: “one supervisor who oversaw the data collection process”

- Line 197-202, what were the convenient locations? How long were the interviews held for/on average how long were interviews?

- Lines 203-211, it is not necessary to write out the dependent and independent variables in this format. They should just be included in the tables and charts associated with the manuscript

- Line 233, there should be a comma between quality and pre-test

- Line 268, what constitutes poor social support? How was this being defined for participants? How did participants define their social support as poor?

**Do you want your identity to be public for this peer review?** For information about this choice, including consent withdrawal, please see our Privacy Policy

Reviewer #1: **Yes: ** DR. JOSEPH KOFI AIDOO

Reviewer #2: No

Reviewer #3: No

---

## [Editor Report · Decision Letter 1]

22 Apr 2025

Quality of Life among Leprosy Patients: An Analysis of associated Factors

PMEN-D-24-00585R1

Dear Dr Edin,

We are pleased to inform you that your manuscript 'Quality of Life among Leprosy Patients: An Analysis of associated Factors' has been provisionally accepted for publication in PLOS Mental Health.

Best regards,

Kizito Omona, PhD

Academic Editor

PLOS Mental Health

I can see that the comments from the reviewers have been sufficiently addressed